# Integrated Multi-Omics Investigations of Metalloproteinases in Colon Cancer: Focus on MMP2 and MMP9

**DOI:** 10.3390/ijms222212389

**Published:** 2021-11-17

**Authors:** Miriam Buttacavoli, Gianluca Di Cara, Elena Roz, Ida Pucci-Minafra, Salvatore Feo, Patrizia Cancemi

**Affiliations:** 1Department of Biological Chemical and Pharmaceutical Sciences and Technologies (STEBICEF), University of Palermo, 90128 Palermo, Italy; miriam.buttacavoli@unipa.it (M.B.); lucadicar@gmail.com (G.D.C.); salvatore.feo@unipa.it (S.F.); 2La Maddalena Hospital III Level Oncological Department, Via San Lorenzo Colli, 90145 Palermo, Italy; roz@lamaddalenanet.it; 3Experimental Center of Onco Biology (COBS), 90145 Palermo, Italy; pucci.ida@gmail.com

**Keywords:** matrix metalloproteinases, gene expression, colon cancer, proteomics, bioinformatics, MMP2, MMP9, functional analysis

## Abstract

Colorectal cancer (CRC) develops by genetic and epigenetic alterations. However, the molecular mechanisms underlying metastatic dissemination remain unclear and could benefit from multi-omics investigations of specific protein families. Matrix metalloproteinases (MMPs) are proteolytic enzymes involved in ECM remodeling and the processing of bioactive molecules. Increased MMP expression promotes the hallmarks of tumor progression, including angiogenesis, invasion, and metastasis, and is correlated with a shortened survival. Nevertheless, the collective role and the possible coordination of MMP members in CRC are poorly investigated. Here, we performed a multi-omics analysis of MMP expression in CRC using data mining and experimental investigations. Several databases were used to deeply mine different expressions between tumor and normal tissues, the genetic and epigenetic alterations, the prognostic value as well as the interrelationships with tumor immune-infiltrating cells (TIICs). A special focus was placed on to MMP2 and MMP9: their expression was correlated with immune markers and the interaction network of co-expressed genes disclosed their implication in epithelial to mesenchymal transition (EMT) and immune response. Finally, the activity levels of MMP2 and MMP9 in a cohort of colon cancer samples, including tissues and the corresponding sera, was also investigated by zymography. Our findings suggested that MMPs could have a high potency, as they are targeted in colon cancer, and might serve as novel biomarkers, especially for their involvement in the immune response. However, further studies are needed to explore the detailed biological functions and molecular mechanisms of MMPs in CRC, also in consideration of their expression and different regulation in several tissues.

## 1. Introduction

Colorectal cancer (CRC) represents the third most common type of cancer and the second most prevalent cause of cancer death, and its incidence is increasing [1]. Colorectal cancer develops from the accumulation of somatic genetic alterations. Fearon and Vogelstein proposed a multi-step model of carcinogenesis, in which accumulating mutations occur early during carcinogenesis; concurrently, additional mutations ensure the full transition to malignancy [2]. This pathway is also called the chromosomal instability (CIN) pathway. Another pathway for CRC pathogenesis involves microsatellite instability [3]. Cumulatively, these genetic changes ensure the tumor’s anti-apoptotic, pro-angiogenic and proliferative properties, consequential to the altered expression of oncogenes, tumor suppressor genes, and/or microRNAs and non-coding RNAs, which contribute to tumor progression. However, colorectal carcinogenesis is a more complicated process and often results from a complex crosstalk between several mutational pathways that explain, in part, the biological, clinical and outcome heterogeneity between patients [4]. For instance, it is well known that outcomes depend on the stage at which the disease is detected: CRC is potentially curable if detected early before the development of metastasis. In later stages, approximately 60–70% of patients with primary colorectal cancer develop metastases, especially in the liver [5]. Thus, the identification of molecular processes underlying tumor progression or the acquired ability to metastasize is necessary to better understand colon tumor biology. The discovery of new biomarkers able to detect colon cancer at early stages and/or that can differentiate metastatic versus nonmetastatic cancers is necessary both for an accurate diagnosis and the development of new targeted therapies.

Matrix metalloproteinases (MMPs) are able to degrade virtually all the components of the extracellular matrix (ECM), collagens, laminin, fibronectin, vitronectin, entactin, and proteoglycans. As a major component of the tumor microenvironment, the ECM exerts dominant roles during tumor progression. In addition, ECM proteins by binding cell surface receptors or non-canonical growth factor presentation provide both biophysical and biochemical cues that are major regulators of cellular behavior [6]. The proteolytic activity of MMPs influences cell proliferation, migration, and adhesion, as well as tissue remodeling, such as angiogenesis, bone development, wound healing, and inflammation [7]. Depending on the context, the processing of ECM components by MMPs can yield bioactive fragments or release non-covalently bound growth factors, chemokines, and adhesion molecules, so they can either promote or suppress several biological functions [8]. So far, 24 MMPs have been reported. Based on their substrate specificity, MMPs are further classified into collagenases (MMP1, MMP8, and MMP13), gelatinases (MMP2 and MMP9), stromelysins (MMP3, MMP10, and MMP11), matrilysins (MMP7 and MMP26), membrane-bound/associated MMPs (MMP14 to MMP17, MMP24, and MMP25) and others (MMP12, MMP19 to MMP23, MMP27, and MMP28) [9]. MMP’s activity is regulated at different levels that include the modulation of gene expression and post-transcriptional regulation, pro-enzyme activation, inhibition by protein inhibitors, and compartmentalization. In fact, most MMPs are induced after cytokine and growth factor stimulation and secreted as inactive zymogens and activated in the extracellular space. Moreover, in the extracellular space, several MMPs can be inhibited by their endogenous tissue inhibitors, the TIMPs (tissue inhibitors of metalloproteinases). Four different forms (TIMP-1 to TIMP-4) have been described to date [10]. Finally, MMPs can be stored in inflammatory cell granules, and released upon cell stimulation. MMPs are deregulated in almost every type of cancer and their expression is linked to a variety of aspects of cancer progression, including proliferation, invasion, EMT, metastasis, and angiogenesis [11]. MMPs are often associated with a poor prognosis in several cancer types [12,13,14]. In colon cancer, the expression of MMPs increases along with tumor progression and some MMPs correlate with tumor invasion, metastasis, or poorer outcomes [15,16,17,18], although some MMPs have been shown to perform tumor-protective effects. However, the collective role of MMPs in different key aspects of colon progression is not yet fully explored. Here, we performed a systematic multi-omics analysis of MMP expression in CRC, with a special focus on MMP2 and MMP9 for their central roles in invasion and metastases. We investigated the expression level of MMP members and determined their correlation with prognosis as well as the interrelationships with tumor immune-infiltrating cells (TIICs), based on the data mining performed using several databases such as Oncomine, GEPIA, Colonomics, TIMER, and UCSC Xena. CBioportal and UALCAN were used to assess the genetic and epigenetic alterations of MMP members. The results underlined the prominent role of MMPs as oncogenes and in some cases as onco-suppressor genes as well as their correlation with immune-infiltrating cells. Moreover, we showed that MMP2 and MMP9 expression was correlated with the immune markers and the interaction network of their co-expressed genes was connected with epithelial to mesenchymal transition (EMT) and the immune response. Finally, the activity levels of both MMP2 and MMP9 were investigated by zymography in 26 colon cancer tissues and paired normal tissues, as well as in the corresponding sera. Both gelatinases were significantly upregulated in colon cancer tissues compared to normal adjacent tissues. No correlation was found between MMP expression in the tissues and sera, indicating the complex regulation of these important enzymes in colon cancer tumorigenesis. Altogether, the obtained results showed that MMPs could have a high potency as a target in colon cancer and might serve as novel biomarkers, especially in consideration of their involvement in the immune response. However, further studies are needed to explore the detailed biological functions and molecular mechanisms through which they act in colon tumor progression.

## 2. Results

We analyzed by data mining 24 MMP genes, grouped into six sub-families according to their substrate specificity (Table 1):(1)Collagenases: MMP1, MMP8, and MMP13;(2)Gelatinases: MMP2 and MMP9;(3)Stromelysins: MMP3, MMP10, and MMP11;(4)Matrilysins: MMP7 and MMP26;(5)Membrane-bound/associated MMPs: MMP14 to MMP17, MMP24, and MMP25;(6)Others MMPs: MMP12, MMP19 to MMP23, MMP27, and MMP28.

### 2.1. Differential Expression of MMP Members between Normal and Cancer Tissues

The MMP expression levels between tumor and normal tissues were analyzed in different cancer types using the Oncomine database (Appendix A). The database performed a total of 9019 unique analyses for all the MMP genes across a wide variety of datasets in different cancer types and 561 showed a significant statistical difference for mRNA expression. In particular, MMP members were found to be upregulated in tumoral tissues in 424 analyses and downregulated in 137 analyses. In more detail, the MMPs were shown to be deregulated (both upregulated and downregulated), especially in colon cancer, followed by breast cancer and leukemia (Figure 1A,C). Interestingly, while the MMPs members from MMP1 to MMP14 were generally upregulated in several cancer types, the others (MMP15 to MMP28) were downregulated (Figure 1B,D). The deregulated expression of MMP members in CRC compared to normal tissues was also validated in the UALCAN and Colonomics databases. The differences between normal and tumor tissues were considered significant when *p* < 0.01. Interestingly, 14 of 24 MMP members were soundly deregulated (Table 2); among these, MMP1, MMP3, MMP7, MMP9, MMP10, MMP11, MMP12, MMP13, and MMP14 were significantly upregulated in tumor tissues, while MMP15, MMP17, MMP25, MMP27, and MMP28 were significantly downregulated in CRC compared with normal tissues. Other MMP members, namely MMP2, MMP8, MMP20, and MMP23B, showed more variability among different databases. Collectively, these results suggest that albeit with a certain variability, the MMP family members perform important roles in CRC, acting essentially as oncogenes but in some cases as onco-suppressor genes. We further analyzed the relationship between MMPs and clinical stages using UALCAN. Collectively, significant differences between normal and patients were recorded, while generally MMP expression was not significantly correlated with cancer stage (Appendix A).

### 2.2. Genetic and Epigenetic Alterations of MMP Members

To investigate the potential factors that may affect MMP expression, we explored the association of MMP expression levels with genetic alterations and promotor methylation status using cBioPortal and UALCAN databases. Interestingly, the mutation rate of MMPs did not exceed 2.5% except for MMP9, whose mutation rate was 11%, with amplification as the main alteration (Figure 2A). Concerning promoter methylation status, all the differentially expressed MMPs showed statistical differences in promoter methylation between normal and colon cancer samples, except for MMP10 (Figure 2B). According to the expression pattern analysis, all the MMPs upregulated in CRC showed lower promoter methylation, except MMP11, while all the MMPs downregulated in CRC showed higher promoter methylation in tumor samples compared with normal samples, except for MMP15 and MMP27. Based on beta values (which indicate hypermethylation or hypomethylation), 5 MMPs were found with the hypermethylated promoter (beta value: 0.7–0.5), while 10 MMPs showed a hypomethylated promoter (beta-value: 0.3–0.25). The obtained results indicate that the deregulated MMP gene expression associated with CRC may be partly due to genetic and epigenetic alterations, but these analyzed alterations cannot fully explain the deregulation mechanisms that, therefore, should be further investigated.

### 2.3. Prognostic Value of MMP Expression in CRC

Next, the relationship between the MMP expression and prognosis in CRC was investigated. The survival outcome was evaluated as overall survival (OS) in GEPIA, UALCAN, Colonomics, and UCSC Xena databases, and disease-free survival (DFS) when available. For each searched gene, patients were divided into two groups (high and low), based on the median expression levels on individual MMPs. Differences in MMP expression were considered significantly associated with prognosis when *p* < 0.05 (Table 3). Interestingly, a pronounced divergence between different databases was recorded. However, consistent results were obtained regarding the association with prognosis for several MMPs members; in particular, high expression levels of MMP2, MMP8, MMP14, MMP16, MMP17, MMP19, and MMP23B were associated with a worse prognosis, while high expression levels of MMP1, MMP3, MMP10, MMP12, and MMP25 were associated with a better prognosis. MMP7, MMP9, MMP11, MMP13, MMP20, MMP21, MMP3A, MMP24, and MMP26-28 did not associate with prognosis in any database. Since the MMPs are expressed by epithelial tumoral cells but also by stromal and the infiltrating cells, we surmise that the observed heterogeneity could be dependent on the different content of tumoral cells within the tissues or the post-transcriptional/translational regulation.

### 2.4. Correlation between MMPs and Immune Infiltration Levels in CRC

Immune infiltrates in cancer cells are almost heterogeneous in patients with the same tumor type and may impact clinical outcomes. The TIMER database was employed to determine whether MMP expression was associated with the tumor purity (e.g., the percentage of tumor cells in the tumor tissue) and the abundance of immune cell subsets, including B cells, T cells, macrophages, neutrophils, and dendritic cells (Appendix A, Table 4). Genes highly expressed in the microenvironment are expected to have negative associations with tumor purity, while the opposite is expected for genes highly expressed in the tumor cells. Interestingly, all the MMPs showed a negative association with tumor purity, except for MMP15, MMP20, MMP23A, MMP24, and MMP26, for which no significant association was recorded. Notably, most of the MMPs shared the same immune cell profile, showing a clear association with dendritic cells, neutrophils, macrophages, and to a lesser extent with CD4+ T and CD8+ T populations cells. On the other hand, few MMPs showed a significant association with B cells. Additionally, stronger correlations between MMPs and immune infiltration were detected for MMP2, MMP9, MMP11, MMP12, and MMP14. Based on the findings in immune infiltration landscapes, we further investigated the role of MMP expression and immune infiltration on clinical prognosis; interestingly, we found that the expression levels of all MMP members were associated with survival in higher clinical stages and with infiltrating levels of CD8 T cells (Appendix A). Overall, these results suggest that elevated MMP expression could affect the immune response in CRC.

### 2.5. Correlation Analysis between MMP2 and MMP9 Expression and Immune Marker Sets

Several MMP members, in particular gelatinases, have been described as a “master switch” triggering tumor spread and metastatic progression. MMP2 and MMP9 degrade type IV collagen in basement membranes, increasing tumor aggressiveness. We explored the links between MMP2 and MMP9 expression and related immunological markers in colon cancer by TIMER (Table 5). In detail, the immune markers included CD8+ T cells, T cells (general), B cells, monocytes, TAMs, M1 and M2 macrophages, neutrophils, NK cells, DCs, Th1 cells, Th2 cells, Tfh cells, Th17 cells, Tregs, exhausted T cells, myeloid-derived suppressors cells, and complement proteins. Interestingly, both MMP2 and MMP9 showed similar relationships with immune gene markers; in particular, MMP2 was significantly correlated with 64 out of 70, while MMP9 was significantly correlated with 63 out of 70 immune gene markers. After purity adjustment, due to its influences on the immune infiltration analysis, an inverse correlation between MMP2 and MMP9 and 60 gene marker sets was recorded. The expression levels of most marker set of monocytes (CSF1R, CCL2), TAMs (CD68, IL10), M2 Macrophages (VSIG4, MS4A4A), Dendritic cells (HLA-DPA1, NRP1, ITGAX), Tfh (BCL6), Th17 (FOXP3, CCR8), T reg (TGFB1, CTLA4), T cell exhaustion (HAVCR2, CD14, FERMT3, GPSM3), Myeloid-derived suppressor cells (IL18BP, PSAP), and complement markers (C3, C1S) were found to be strongly correlated with both MMP2 and MMP9. These findings further suggest that MMP2 and MMP9 play pivotal roles in cancer immune escape and support the hypothesis that they could be a novel predictor for immunotherapy response.

### 2.6. MMP2 and MMP9-Gene Co-Expression Networks

The predicted genes transcriptionally co-expressed with MMP2 and MMP9 were retrieved by UALCAN, using a Pearson correlation score of ≥0.4 (Appendix A). Functional enrichments in the MMP2- and MMP9-regulated networks (biological pathways) were highlighted by using the Fun Rich tool. Interestingly, both MMP2 and MMP9 co-expressed genes were involved in similar biological processes (Figure 3A,B), including the epithelial to mesenchymal transition (EMT), integrin family cell-surface interactions, and Beta 1 integrin cell surface interactions. The functional biological pathways enriched from common MMP2/MMP9 co-expressed genes (Figure 3C) included EMT, immune system, cell surface interactions at the vascular wall, thromboxane A2 receptor signaling, and adaptative immune system (Figure 3D). Collectively, the pathway analysis of MMP2/MMP9 co-expressed genes showed their involvement in EMT and Immune system pathways, emphasizing their importance in tumor progression, although the mRNA expression levels were not directly correlated with prognosis.

### 2.7. Activity Levels of Gelatinases MMP9 and MMP2 in Colon Cancer Tissues and Corresponding Sera Samples

Numerous studies have demonstrated increased MMP2 and MMP9 mRNA, protein, and/or activity in colorectal tumor specimens compared to the normal colonic mucosa. The enzymatic activities of MMP2 and MMP9 were evaluated by zymography in a cohort of 26 surgical colon cancer tissues paired with non-tumoral adjacent tissue. As shown in Figure 4A, the majority of the colon cancer tissues showed lytic activities corresponding to the latent and activated forms of MMP2 and MMP9. On the contrary, the non-tumoral adjacent tissues were generally positive only for the latent forms. Moreover, lytic bands of high molecular weight, corresponding to the homodimeric forms of MMP-9 and complexes of pro-MMP9/TIMP1, respectively, also occurred in some samples. In some samples, a lytic band of low molecular weight was also evident, but not well characterized. Further, all non-tumoral adjacent tissues showed very low activity levels of both pro-MMP9 and pro-MMP2, whereas higher activity levels were detected in the paired tumor samples. Interestingly, a high variability of MMP2 and MMP9 activity levels was recorded between different patients, suggesting their possible role in tumor progression. The zymographic analysis was also extended to the sera samples of the same cohort of patients (Figure 4B) collected before surgery. As expected, in all the analyzed samples, only the enzymatic activity of Pro-MMP9 and Pro-MMP2 was detected, while non-enzymatic activity was recorded for the activated forms. The sera samples (*n* = 20) of healthy donors were used as the control (Figure 4C). A densitometric quantification of Pro-MMP9 and Pro-MMP2 in all analyzed samples (Figure 4D) confirmed statistical differences between colon cancer tissues and paired normal tissues. Moreover, the activity levels of Pro-MMP2 were significantly higher in colon cancer sera compared to healthy sera. No significant differences were recorded for Pro-MMP9. Finally, no correlation between colon cancer tissues and sera was detected, suggesting complex regulatory mechanisms of MMPs in different tissues.

## 3. Discussion

The MMP superfamily plays critical roles in ECM remodeling, the activation of cell surface proteins, and the shedding of membrane-bound receptor molecules. MMPs are also involved in a range of biological activities such as cell proliferation, differentiation, tissue morphogenesis, migration, and survival [1]. Often, MMPs act together in a cascading fashion manner [2]. For example, MMP3 and MMP10 activate MMP1, MMP7, MMP8, and MMP9, whereas MMP14 activates both MMP2 and MMP13 in the presence of TIMP2. MMP2 and MMP13 were shown to aid the cleavage of the pro-domain in pro-MMP9 [3]. Since the ECM composition depends on the complex network of MMP cross-activation, if their tight regulation is compromised, the ECM is impaired. For these reasons, MMPs have been increasingly implicated in carcinogenesis. Initially, MMPs were recognized for their role in late-stage tumor progression. However, now it is well known that MMPs deregulation causes an imbalance in crucial signal pathways contributing to the cancer hallmarks, sustaining proliferative signaling, the evasion of growth suppressors, and cell death resistance, which enable replicative immortality. The MMP superfamily is therefore of interest as a potential source of biomarkers and/or targets for therapeutic interventions. Although single MMP members are extensively studied, their collective expression pattern and potential roles in CRC cancer, as well as their underlying regulatory mechanisms, remain poorly investigated. By data mining, we found that collectively MMPs were strongly deregulated in CRC and exert both pro- and anti-tumorigenic effects. In particular, the expression levels of MMP1, MMP3, MMP7, MMP9, MMP10, MMP11, MMP12, MMP13, and MMP14 were significantly upregulated in CRC tumor tissues compared to normal tissues, while MMP15, MMP17, MMP25, MMP27, and MMP28 were significantly downregulated. Interestingly, among the MMPs downregulated in CRC, three of them are membrane-type MMPs (MT-MMPs). MMP17 and MMP25 are anchored to the plasma membrane via the glycosyl-phosphatidyl inositol (GPI) anchor, conferring them with a unique set of regulatory and functional mechanisms [4]. GPI-MT-MMP’s function and regulation is still not fully understood [5], and future studies need to address their biosynthetic pathway, endocytosis, recycling, shedding, and the function and significance of dimerization/oligomerization. According to our findings, MMP-15 is an anti-apoptotic factor and a prognostic predictor for disease-free survival in colorectal cancer [6]. Among the upregulated MMPs, the promoters of MMP1, MMP3, MMP7, MMP9, MMP10, MMP12, and MMP13 have a TATA box and AP-1 binding sites [7], consistent with observations that some MMPs are co-regulated in their expression. Since several cytokines or growth factors, well recognized as prognostic factors for CRC, including platelet-derived growth factor (PDGF) [8], tumor necrosis factor-α (TNF-α) [9], interleukins (ILs) [10], epidermal growth factor (EGF) [11], vascular endothelial growth factor (VEGF) [12] basic fibroblast growth factor (bFGF) [13,14] and the TGFβ [15] interact with TATA box and AP-1 elements, we surmise that they can participate in deregulation of MMP expression in CRC. Surprisingly, among the upregulated MMPs, MMP-2 did not show the inclusion criteria to consider it as upregulated in CRC, infact only in the UALCAN database, MMP2 was significantly upregulated in CRC compared to normal adjacent tissues. This finding underlined the complexity of the MMP expression and activity, which depend on tightly regulatory mechanisms, activation of the zymogen precursors by specific proteases and cytokines present in the milieu, location inside or outside the cell, interaction with specific components of the ECM, internalization, and inhibition by tissue inhibitors of metalloproteinases (TIMPs). Unlike classical oncogenes, generally, the MMPs are not upregulated by gene amplification or mutation. The increased MMP expression in CRC is mainly due to transcriptional changes and/or epigenetic modifications [16,17]. Cytosine methylation within CpG in the promoter region inhibits gene expression, and it was suggested that hypomethylation regulates MMP expression [18]. All the differentially expressed MMPs showed statistical differences in promoter methylation between normal and colon cancer samples with a general tendency of hypomethylated promoters for upregulated MMPs and hypermethylated promoters for downregulated MMPs, while their mutation rate did not exceed 2.5%, except for MMP9, whose mutation rate was 11%. Nevertheless, further studies are needed to more comprehensively explore the detailed molecular mechanisms of altered MMP expression in CRC.

When the prognostic value of MMP expression was assessed, a pronounced divergence between different databases was recorded, probably because the cellular source of a specific MMP could impact the biological outcome associated with its expression. The dynamic nature of the tumor microenvironment during cancer progression affects the availability of substrate repertoire as well as the expression of various MMPs. The tumor microenvironment (TME) is an ecosystem composed of immune cells, stromal cells, tumor cells, and secreted cytokines and chemokines [19,20]. The TME has been reported to regulate cancer initiation and progression and is valuable in prognosis. Additionally, immune cells, as important components of the TME, have been proved to be closely associated with the clinical outcome of cancer [21]. In situ hybridization techniques have shown that most MMPs are predominantly produced by adjacent host stromal and inflammatory cells in response to factors released by tumors [22]. In this study, we demonstrated that MMP expression was closely correlated with immune infiltration, especially with dendritic cells, neutrophils, macrophages and to a lesser extent with CD4+ T and CD8+ T populations cells. Stronger correlations between MMPs and immune infiltration were detected for MMP2, MMP9, MMP11, MMP12, and MMP14. During inflammation, neutrophils synthesize MMP8 and MMP9, which are stored in granules before activation [23]. Macrophages secrete MMP1, MMP2, MMP3, MMP7, and MMP9, as well as MMP12. T-Cells secrete MMP2 and MMP9. MMP-9 is overexpressed by endothelial cells and myeloid cells [24]. Our results suggest that the expression of MMPs is regulated in a paracrine manner by inflammatory cells as well as by tumor or stromal cells, with a continuous bidirectional crosstalk. The interactive circuits between tumor cells and the microenvironment are difficult to predict a priori, but influence the milieu, leading to the rapid and critical response required for tumor progression. Greater knowledge of the timely alteration of cellular environments by MMPs will be of great value not only for a better understanding the basic biology of the matrix and its turnover, but also for insights into the underlying mechanisms of immune system dysregulation, which will be useful for clinical applications, especially for immunotherapy. Due to their capacity to degrade basement membrane components, the gelatinases MMP2 and MMP9 are key molecules for the spreading of tumor and metastatic progression. We explored the links between MMP2 and MMP9 expression and related immunological markers in CRC. The immune cells included CD8+ T cells, T cells (general), B cells, monocytes, TAMs, M1 and M2 macrophages, neutrophils, NK cells, DCs, Th1 cells, Th2 cells, Tfh cells, Th17 cells, Tregs, exhausted T cells, myeloid-derived suppressors cells, and complement proteins. Interestingly, both MMP2 and MMP9 showed similar relationships with immune gene markers. The expression levels of most marker set of monocytes (CSF1R, CCL2), TAMs (CD68, IL10), M2 macrophages (VSIG4, MS4A4A), dendritic cells (HLA-DPA1, NRP1, ITGAX), Tfh (BCL6), Th17 (FOXP3, CCR8), T reg (TGFB1, CTLA4), T cell exhaustion (HAVCR2, CD14, FERMT3, GPSM3), myeloid-derived suppressor cells (IL18BP, PSAP), and complement markers (C3, C1S) were found to be strongly correlated with both MMP2 and MMP9. These findings further suggest that MMP2 and MMP9 play pivotal roles in cancer immune escape and support the hypothesis that they could be novel predictor for immunotherapy response. Interestingly, the pathway analysis performed on the MMP2 and MMP9 co-expressed genes sustained their involvement in immune system pathways but also in the epithelial to mesenchymal transition (EMT). The EMT represents a hallmark of initial tumorigenesis, where epithelial cells, intimately connected through cell–cell interactions (including tight junctions, adherent junctions, and desmosomes), start to disassemble. This results in a redistribution of transmembrane proteins between epithelial cells, and plaque scaffolding proteins between cells and the actin cytoskeleton. These events are accompanied by the acquisition of a more motile phenotype typical of mesenchymal phenotypes, characterized by enhanced migratory capacity, invasiveness, and increased production of ECM components. MMP9 functions as an inducer of EMT through its proteolytic action on E-cadherin and concomitant loss of other epithelial markers such as vimentin and Snail. On the other hand, E-cadherin inhibition upregulates MMP2, whose expression also depends on HIF-1*α*, involved in the hypoxia response [25,26]. MMP2 and MMP9 mRNA, protein, and/or activity have been found increased in colorectal tumor specimens compared to normal colonic mucosa [27,28,29,30]. In particular, the level of active MMP2 associated with tumor cells located within the invasive regions of colon cancer specimens was increased 10-fold over the level of active MMP2 associated with adjacent normal epithelium within the same tumor specimens [31]. Here, we showed that the enzymatic activities of MMP2 and MMP9 were very low in normal tissues, while higher levels of both pro- and active forms were detected in the paired tumor samples of 24 patients. Interestingly, a high variability of MMP2 and MMP9 activity levels was recorded between different patients, suggesting their possible role in tumor progression. Due to the limited number of analyzed patients, we failed to demonstrate a significant relationship between gelatinase activity and prognosis. The zymographic analysis was also extended to the sera samples of the same cohort of patients, although no correlation between colon cancer tissues and sera was found, suggesting the complex regulatory mechanisms of MMPs in different tissues. Clearly, more research is required to understand this important aspect of MMP pathobiology.

## 4. Materials and Methods

### 4.1. Expression Analysis of MMP Members Using Oncomine, UALCAN, and Colonomics

The differences in mRNA levels of each MMP member between the colon cancer and control samples were evaluated using gene expression array datasets from Oncomine (https://www.oncomine.org/resource/login.html, accessed on 15 March 2021), UALCAN (https://ualcan.path.uab.edu/, accessed on 15 March 2021), and Colonomics (https://www.colonomics.org/expression-browser/, accessed on 15 March 2021), which are public cancer microarray databases, free online accessible. In each database, the analysis was repeated for each MMP member.

#### 4.1.1. Oncomine Database

The Oncomine database [32] analyzes the mRNA expression differences between tumor and normal tissues in 20 different human cancers. The threshold was set according to the following values: *p*-value: 0.01; fold change: 2; gene rank: 10%; analysis type: cancer vs. normal analysis; data type: mRNA [33,34].

#### 4.1.2. UALCAN Database

UALCAN evaluates cancer OMICS data (TCGA, MET500, and CPTAC) [35]. For each search, the database provides graphs and plots depicting the expression profile of protein and miRNA between normal and cancer tissue and evaluates the epigenetic regulation of gene expression by promoter methylation. The expression level of the searched gene is normalized as transcript per million reads, and a *p*-value <0.01 is considered to be significant.

#### 4.1.3. Colonomics Database

Colonomics is a web resource analyzing biomarkers of diagnosis and prognosis in colorectal cancer [36]. For each search, the database provides plots of the gene expression profiles between 98 paired normal and tumor samples, and a *p*-value < 0.01 is considered to be significant.

### 4.2. Genetic Alterations and Epigenetic Regulation of MMP Members Using cBioPortal and UALCAN

The frequency of MMPs alterations (amplification, deep deletion, and missense mutations) in CRC patients was assessed using the OncoPrint tool of cBioPortal (http://www.cbioportal.org, accessed on 15 March 2021). cBioportal is an interactive open-source platform that provides large-scale cancer genomics datasets [37]. Promoter methylation status was analyzed using UALCAN.

### 4.3. Evaluation of the Prognostic Value of MMPs Using GEPIA, UALCAN, Colonomics, and UCSC Xena

The correlation between MMP expression levels and overall survival (OS) or disease-free survival (DFS) in CRC was analyzed by data mining in the GEPIA (http://gepia.cancer-pku.cn/, accessed on 15 March 2021), UALCAN, Colonomics, and UCSC Xena Browser (https://xenabrowser.net/heatmap/, accessed on 15 March 2021). GEPIA (Gene Expression Profiling Interactive Analysis) [38] provides key interactive and customizable functions, including differential expression analysis, profiling plotting, correlation analysis, patient survival analysis, similar gene detection, and dimensionality reduction analysis.

UCSC Xena allows users to explore functional genomic data sets for correlations between genomic and/or phenotypic variables [39]. Each database provided a Kaplan–Meier plot for the searched gene. A *p*-value <0.01 was considered significant.

### 4.4. Association between MMP Expression and Immune Infiltration by TIMER

TIMER is a comprehensive resource for systematical analysis of immune infiltrates across diverse cancer types [40]. The web server consists of different modules: the gene module to explore the correlation between gene expression and abundance of immune infiltrates (B cells, CD4+ T cells, CD8+ T cells, neutrophils, macrophages, and dendritic cells); the survival module to explore the association between clinical outcome and abundance of immune infiltrates or gene expression; the diff Exp module to explore differential gene expression between tumor and normal tissue; correlation module to explore correlations between genes.. The database uses statistical methods, validated by pathological examination, to assess the immune infiltration of tumors by specific immune cell subpopulations (neutrophils, macrophages, dendritic cells, B cells, and CD4/CD8 T cells). Kaplan–Meier curve analysis was also performed to investigate changes in the survival of patients with different levels of gene expression or immune cell infiltration. The correlation of MMP expression with specific immune-infiltrating cell subset markers was also evaluated. These representative gene markers were referenced in the CellMarker database (http://biocc.hrbmu.edu.cn/CellMarker/, accessed on 15 April 2021).

### 4.5. MMP2 and MMP9 Co-Expressed Genes and Pathways Enrichment Analysis

The co-expressed genes related to MMP2 and MMP9 were retrieved using the UALCAN database using Pearson’s correlation coefficient (*p* ≥ 0.4). The gene lists were submitted into the FunRich database [41], (http://www.funrich.org/, accessed on 15 March 2021) a stand-alone software tool used mainly for functional enrichment and interaction network analysis. Enriched biological pathways were ranked by *p*-value and the top six significant pathways were exhibited as bar charts. A *p*-value <0.05 was regarded as statistically significant.

### 4.6. Tissue and Sera Samples from CRC Patients

A total of 26 patients who underwent surgical removal of CRC tumor at La Maddalena Hospital were enrolled for this study. Inclusion criteria were: an age of less than 65 years and no metastases at diagnosis. Patients did not receive any cytotoxic treatment before surgery. The researchers were blinded to clinical characteristics during the analyses. Tumor specimens (CCT) and normal adjacent samples (NAT) were immediately stored at −80 °C until protein extraction [42]. Corresponding blood samples were collected before surgery. The study was carried out with the informed consent of patients and with the approval of the Institutional Review Board (N°515/2008, 13 May 2008) from the La Maddalena Hospital. Healthy sera (*n* = 20) were retrieved from healthy volunteers [43].

After homogenization with 50 mM Tris-HCl pH 7.5, 0.003% penicillin, 0.005% streptomycin, tissue lysates were incubated under rotation overnight at 4 °C [44], and then centrifuged several times at 10,000 rpm for 20 min to remove cell debris. Bradford assay was used to quantify protein content, as already reported [34,45,46,47,48]. Blood, collected into plastic tubes without coagulation accelerators to prevent the release of gelatinases during platelet activation, was centrifuged at 1600× *g* for 10 min 30 min after collection. Sera were aliquoted and used only once to prevent enzymatic activation due to freeze–thawing processes.

### 4.7. Gelatin Zymography

Gelatin zymography assay was performed to detect gelatinase activity (MMP2 and MMP9) after SDS-PAGE electrophoresis separation. Denaturation induced by the SDS-mediated denaturation allow the MMPs enzymatic activation, even if they are with their pro domains. MMP2 (72 kDa) and MMP9 (92 kDa) can be detected on gelatin zymograms as two-three white bands (pro and active forms) after staining with Coomassie Blue, based on their molecular mass. Aliquots of 18 µg of cell lysates from CRC tissues and paired non-tumoral adjacent tissues or 10 µL of sera previously diluted 1:25 were separated by electrophoresis on a 7.5% sodium dodecyl sulfate (SDS)–polyacrylamide gel containing 0.1% gelatin, under non-reducing conditions [49,50,51,52]. After electrophoresis, the gels were washed for 1 h with a buffer containing 50 mM Tris-HCl, pH 7.5 and 2.5% Triton X-100, and then incubated for 18 h at 37 °C with 50 mM Tris-HCl pH 7.5, 150 mM NaCl, 5 mM CaCl_2_. Gels were stained with Coomassie Brilliant Blue G 250 and de-stained with H_2_O milliQ. The band’s intensity was quantified with ImageJ software.

## 5. Conclusions

The information gained from this analysis supports the idea that most MMPs perform complex roles in colon cancer, acting as pro-tumorigenic, anti-tumorigenic, or null factors. Since the MMP expression is regulated on multiple levels, the involvement of distinct MMPs in cancer progression, derived from immune cells, might be more complex than previously expected. Moreover, the hierarchical relationship between individual MMPs and cells, cytokines, and chemokines highlights the need for new multi-omics approaches that take into account the individual genomic, proteomic and metabolomic profiles of colon cancer patients. Moreover, although early efforts of targeting MMPs largely failed in the later stages of clinical trials, a better understanding of the function of MMPs in CRC should address the design of “smart”, MMP-responsive drugs that are useful as therapeutics.

## Figures and Tables

**Figure 1 ijms-22-12389-f001:**
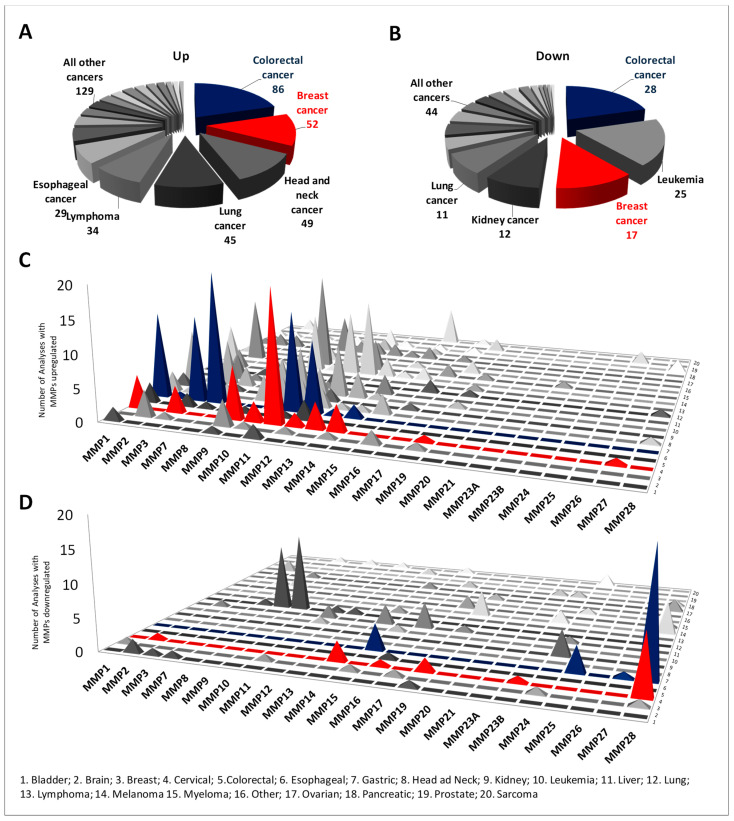
Differential expression of MMP members between normal and cancer tissues in different cancer types using Oncomine. (**A**–**C**) Pie charts showing the number of analyses in which the MMP members were found to be significantly deregulated (upregulated or downregulated) in cancer compared to normal tissues. (**B**–**D**) Histograms showing the number of analyses in which the MMPs were found to be significantly deregulated, sorted for individual MMPs and cancer type.

**Figure 2 ijms-22-12389-f002:**
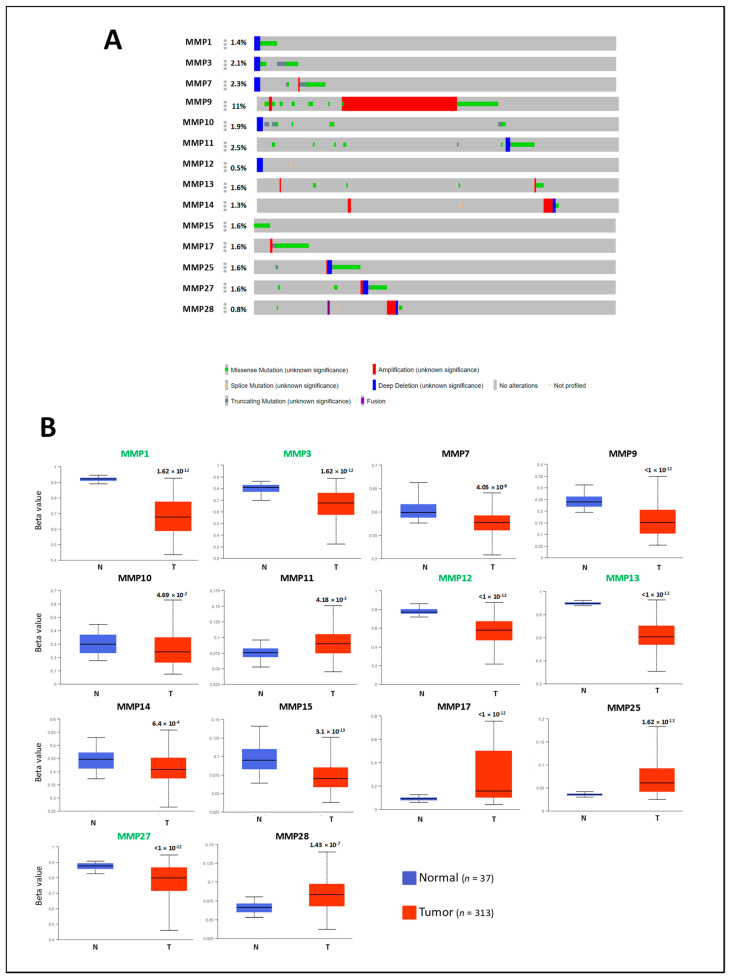
Analysis of genetic and epigenetic alterations of MMP members differentially expressed in CRC. (**A**) The percentage of MMPs alterations was retrieved by using the OncoPrint tool of cBioPortal. Red and blue represent amplification, and deep deletion, respectively; green and grey represent missense and truncating mutations. (**B**) Box plots showing the MMPs promoter methylation status between normal and CRC, retrieved from the UALCAN database. Beta value indicates level of DNA methylation ranging from 0 (un methylated) to 1 (fully methylated). Different Beta value cut-off has been considered to indicate hypermethylation [beta value: 0.7–0.5] or hypomethylation [beta-value: 0.3–0.25]. MMP members with hypermethylated promoters are marked in green; MMP members with hypomethylated promoters are marked in black. *p* <0.01 was considered significant.

**Figure 3 ijms-22-12389-f003:**
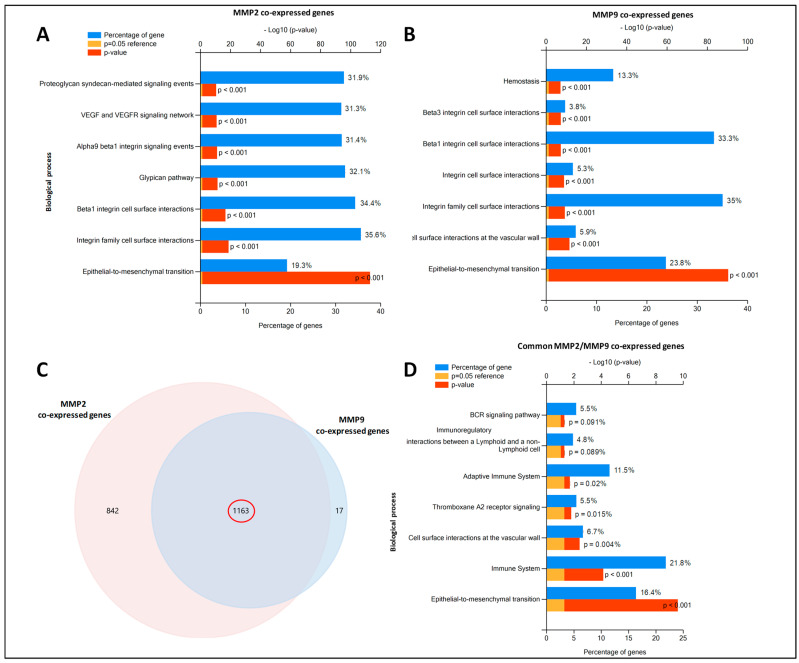
Enrichment pathway analysis of MMP2/MMP9 co-expressed genes retrieved by UALCAN database. (**A**,**B**) Significantly enriched biological pathways were ranked by *p*-value using the FunRich 3.0 software. For each biological pathway, the percentage of proteins and their *p*-values (Benjamini–Hochberg correction) are indicated. Common pathways between MMP2 and MMP9 co-expressed genes are highlighted with different color squares. (**C**) Venn diagram of overlapping MMP2/MMP9 co-expressed genes. (**D**) Significantly enriched biological pathways of overlapping MMP2/MMP9 co-expressed genes, ranked by *p*-value.

**Figure 4 ijms-22-12389-f004:**
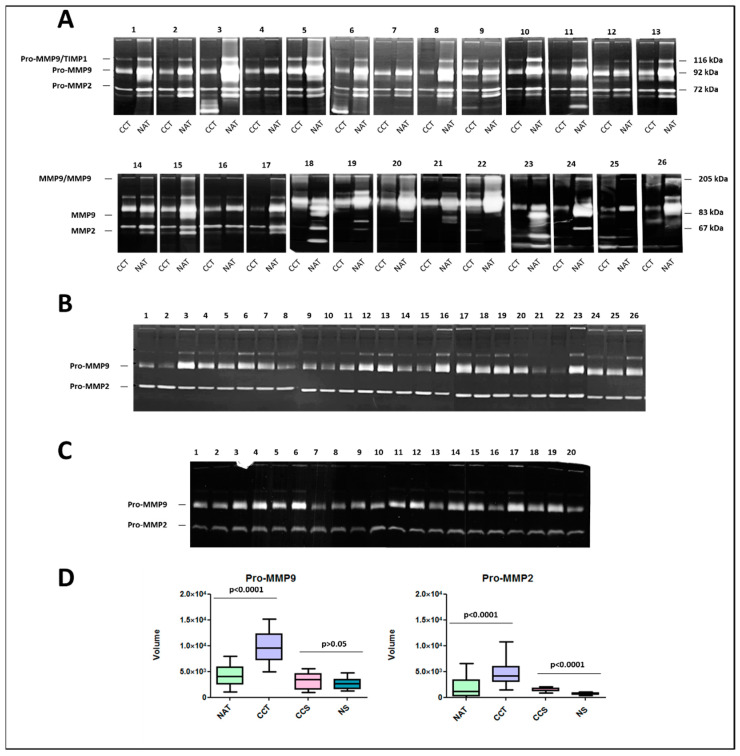
MMP9 and MMP2 activity levels in colon cancer tissues, normal adjacent tissues, and sera. (**A**) Gelatin zymography of protein extracts from 26 colon cancer tissues (CCT) and 26 paired normal adjacent tissues (NAT). (**B**) Gelatin zymography of 26 sera from the same patients, collected before surgery and (**C**) 20 sera from healthy donors. (**D**) Box plot of lytic bands quantification. The experiments were performed in triplicate. *p*-value indicates statistical significance.

**Table 1 ijms-22-12389-t001:** List of 24 MMPs classified into 6 sub-families according to their substrate specificity (collagenases, gelatinases, stromelysins, matrilysins, mmbrane-bound/associated MMPs, and other MMPs). For each MMP, the gene name, chromosomic localization, protein name, uniprot database access number, theoretical molecular weight (*M*_W_) and isoelectric point (p*I*) are reported.

Gene Name	Chromosomic Localization	Protein Name	Uniprot Access Number	*M*_W_ (Da)/p*I*
Collagenases				
MMP1	11q22.3	Interstitial collagenases	P03956	54,061/6.97
MMP8	11q22.3	Neutrophil collagenase	P22894	53,411/6.86
MMP13	11q22.3	Collagenase-3	P45452	53,819/5.31
Gelatinases				
MMP2	16q13	72 kDa type IV collagenase	P08253	73,881/5.09
MMP9	20q11.2	Matrix metalloproteinase-9	P14780	78,457/5.91
Stromelysins				
MMP3	11q22.3	Stromelysin-1	P08254	53,976/6.07
MMP10	11q22.3	Stromelysin-2	P09238	54,150/5.59
MMP11	22q11.2	Stromelysin-3	P24347	54,589/6.87
Matrilysins				
MMP7	11q21	Matrilysin	P09237	29,676/8.09
MMP26	11p15	Matrix metalloproteinase-26	Q9NRE1	29,708/6.45
Membrane-bound/associated MMPs				
MMP14	14q11	Matrix metalloproteinase-14	P50281	65,893/7.87
MMP15	16q13	Matrix metalloproteinase-15	P51511	75,806/7.49
MMP16	8q21	Matrix metalloproteinase-16	P51512	69,521/8.74
MMP17	12q24.3	Matrix metalloproteinase-17	Q9ULZ9	66,652/6.55
MMP24	20q11.2	Matrix metalloproteinase-24	Q9Y5R2	73,230/9.58
MMP25	16p13.3	Matrix metalloproteinase-25	Q9NPA2	62,553/8.76
Others MMPs				
MMP12	11q22.3	Macrophage metalloelastase	P39900	54,001/8.89
MMP19	12q14	Matrix metalloproteinase-19	Q99542	57,356/7.67
MMP20	11q22.3	Matrix metalloproteinase-20	O60882	54,386/9.08
MMP21	10q26.13	Matrix metalloproteinase-21	Q8N119	65,042/9.42
MMP23A	1p36.33	Matrix metalloproteinase-23A	O75900	43,934/9.94
MMP23B	1p36.34	Matrix metalloproteinase-23B	O75901	43,934/10.27
MMP27	11q24	Matrix metalloproteinase-27	Q9H306	59,025/8.95
MMP28	17q11	Matrix metalloproteinase-28	Q9H239	58,938/10.07

**Table 2 ijms-22-12389-t002:** Differential expression analysis of MMP members between normal and colon cancer tissues using UALCAN and Colonomics. Databases were searched for MMP expression between normal and tumoral tissues and differences were considered significant when *p* < 0.01. The MMP members significantly upregulated in tumors compared to normal tissues are: MMP1, MMP3, MMP7, and MMP9-14. The MMP members significantly downregulated in tumors compared to normal tissues are MMP15, MMP17, MMP25, MMP27, and MMP28.

	UALCAN (T = 286; *n* = 41)	Colonomics (T = 98; *n* = 98)
MMP1	*p* = 1.6 × 10^−12^	*p* < 2 × 10^−16^
MMP2	*p* = 2.1 × 10^−3^	-
MMP3	*p* < 1 × 10^−12^	*p* < 2 × 10^−16^
MMP7	*p* = 1.6 × 10^−12^	*p* < 2 × 10^−16^
MMP8	-	*p* = 7.5 × 10^−4^
MMP9	*p* = 1.4 × 10^−10^	*p* = 3.9 × 10^−16^
MMP10	*p* = 1.8 × 10^−6^	*p* < 2 × 10^−16^
MMP11	*p* = 1.6 × 10^−12^	*p* < 2 × 10^−16^
MMP12	*p* = 2.22 × 10^−9^	*p* < 2 × 10^−16^
MMP13	*p* = 6.2 × 10^−9^	*p* = 8.9 × 10^−8^
MMP14	*p* = 1.62 × 10^−12^	*p* < 2 × 10^−16^
MMP15	*p* = 6 × 10^−16^	*p* < 2 × 10^−16^
MMP16	-	-
MMP17	*p* = 1.9 × 10^−5^	*p* = 3.1 × 10^−3^
MMP19	-	-
MMP20	*p* = 3.6 × 10^−5^	-
MMP21	-	-
MMP23A	-	-
MMP23B	*p* = 1.2 × 10^−3^	-
MMP24	-	-
MMP25	*p* = 7.6 × 10^−7^	*p* = 1.9 × 10^−6^
MMP26	-	-
MMP27	*p* = 4.7 × 10^−12^	*p* < 2 × 10^−16^
MMP28	*p* < 1 × 10^−12^	*p* < 2 × 10^−16^

**Table 3 ijms-22-12389-t003:** Prognostic analysis of MMPs in CRC evaluated by GEPIA, Colonomics, UALCAN, and UCSC Xena. The *p*-value was marked in red when higher expression of selected MMP was associated with poor prognosis and in the black when higher expression of selected MMP was associated with a good prognosis. Colon Adenocarcinoma datasets (COAD), Colon and Rectal Adenocarcinoma datasets (COADREAD), Disease-free survival (DFS), and overall survival (OS).

	GEPIA (COAD)		GEPIA (READ)		Colonomics (COAD)		UALCAN (COAD)	UCSC Xena COAD COADREAD READ			
	DFS	OS	DFS	OS	DFS	OS	OS	GDC TCGA OS	TCGA OS	TCGA OS	TCGA OS
MMP1	-	-	-	-	-	-	-	*p* = 0.019	-	-	-
MMP2	*p* = 0.017	-	-	-	-	-	-	-	-	-	-
MMP3	-	-	-	-	-	*p* = 0.046	-	*p* = 0.004	-	-	-
MMP7	-	-	-	-	-	-	-	-	-	-	-
MMP8	*p* = 0.043	-	-	-	-	-	-	-	-	-	-
MMP9	-	-	-	-	-	-	-	-	-	-	-
MMP10	-	-	-	-	-	-	-	*p* = 0.0088	-	-	-
MMP11	-	-	-	-	-	-	-	-	-	-	-
MMP12	-	-	-	-	-	*p* = 0.0087	-	*p* = 0.027	-	*p* = 0.037	-
MMP13	-	-	-	-	-	-	-	-	-	-	-
MMP14	*p* = 0.012	*p* = 0.013	*p* = 0.0037	-	-	-	-	-	-	*p* = 0.046	-
MMP15	-	-	-	-	-	*p* = 0.014	*p* = 0.02	-	-	-	-
MMP16	-	-	-	-	-	-	-	-	*p* = 0.016	*p* = 0.017	-
MMP17	-	-	-	-	-	-	-	-	*p* = 0.011	-	-
MMP19	-	-	-	-	-	-	-	-	*p* = 0.002	*p* = 0.00033	-
MMP20	-	-	-	-	-	-	-	-	-	-	-
MMP21	-	-	-	-	-	-	-	-	-	-	-
MMP23A	-	-	-	-	-	-	-	-	-	-	-
MMP23B	-	*p* = 0.008	-	-	-	-	*p* = 0.0055	-	-	*p* = 0.014	-
MMP24	-	-	-	-	-	-	-	-	-	-	-
MMP25	-	-	-	*p* = 0.029	-	-	-	-	-	-	-
MMP26	-	-	-	-	-	-	-	-	-	-	-
MMP27	-	-	-	-	-	-	-	-	-	-	-
MMP28	-	-	-	-	-	-	-	-	-	-	-

**Table 4 ijms-22-12389-t004:** Correlation analysis between MMPs and immune infiltration levels. The correlations between individual MMPs and tumor purity, MMPs and infiltration levels of six TIICs (B-cells, CD4+ T-cells, CD8+ T-cells, macrophages, neutrophils, and dendritic cells) were analyzed by TIMER in COAD. r, the categorized Pearson’s correlation coefficient, was scored as follows: (---), r from −1.0 to −0.5, strong negative association; (--), −0.5 to −0.3, weak negative association; (-), −0.3 to 0.1, little association. (+), +0.1 to 0.3, little association; (++) +0.3 to +0.5, weak positive association; (+++), +0.5 to +1.0, strong positive association.

	Purity	B Cell	CD8+ T Cell	CD4+ T Cell	Macrophage	Neutrophil	Dendritic Cell
MMP1	-		+		+	+++	++
MMP2	--		+	++	+++	+++	+++
MMP3	-	-	+			++	+
MMP7	-	+				+	
MMP8	-	-	+		++	++	++
MMP9	-		+	++	++	+++	+++
MMP10	-					+	+
MMP11	-		-	++	++		+
MMP12	-	+	++	+	++	+++	+++
MMP13	-		+	+	++	++	++
MMP14	--		+	+++	+++	+++	+++
MMP15			-	+	-		
MMP16	-		+	++	+++	++	++
MMP17	-			++	+	+	+
MMP19	--	+	+	+++	++	++	++
MMP20							
MMP21	-		+	+	+	+	+
MMP23A			-				
MMP23B	-			++	+	+	+
MMP24			-	+		-	-
MMP25	--	+	+	++	+	+++	+++
MMP26				-		-	-
MMP27	-		+	+	+	+	+
MMP28	-		+	+	+	+	+

**Table 5 ijms-22-12389-t005:** Correlation analysis between MMP2 and MMP9 expression and related genes and markers of infiltrating immune cells using the TIMER database. None, correlation without adjustment; purity, correlation adjusted by purity. r, the categorized Pearson’s correlation coefficient, was scored as follows: (---), r from −1.0 to −0.5, strong negative association; (--), −0.5 to −0.3, weak negative association; (-), −0.3 to 0.1, little association. (+), +0.1 to 0.3, little association; (++) +0.3 to +0.5, weak positive association; (+++), +0.5 to +1.0, strong positive association. *p*-value indicates statistical significance.

		MMP2		MMP9	
	Markers	None	Purity	None	Purity
		r	r	r	r
CD8+ T cell	CD8A	(++)	(--)	(++)	(--)
	CD8B	(+)	(-)	(+)	(-)
	CD3D	(++)	(--)	(++)	(--)
T cell (general)	CD3E	(++)	(--)	(++)	(--)
	CD2	(++)	(--)	(++)	(--)
	CD19	(++)	(--)	(+)	(--)
B cell	CD79A	(++)	(--)	(++)	(--)
	CD86	(+++)	(--)	(+++)	(--)
Monocyte	CSF1R	(+++)	(--)	(+++)	(--)
	CCL2	(+++)	(--)	(+++)	(--)
TAM	CD68	(+++)	(--)	(+++)	(--)
	IL10	(+++)	(--)	(++)	(--)
	NOS2		(-)		(-)
M1 macrophage	IRF5	(++)		(++)	
	PTGS2	(++)	(-)	(+)	(-)
	CD163	(+++)	(--)	(+++)	(--)
M2 macrophage	VSIG4	(+++)	(--)	(+++)	(--)
	MS4A4A	(+++)	(--)	(+++)	(--)
	CEACAM8	(-)	(-)		(-)
Neutrophils	ITGAM	(+++)	(--)	(+++)	(--)
	CCR7	(++)	(--)	(++)	(--)
	KIR2DL1	(+)	(-)	(+)	(-)
	KIR2DL3	(+)	(-)	(+)	(-)
	KIR2DL4	(+)	(--)	(+)	(--)
Natural killer cells	KIR3DL1	(+)	(-)	(+)	(-)
	KIR3DL2	(+)	(-)	(+)	(-)
	KIR3DL3			(+)	
	KIR2DS4	(+)	(-)	(+)	(-)
	HLA-DPB1	(+++)	(--)	(+++)	(--)
	HLA-DQB1	(++)	(--)	(++)	(--)
	HLA-DRA	(+++)	(--)	(++)	(--)
Dendritic cell	HLA-DPA1	(+++)	(--)	(+++)	(--)
	CD1C	(++)	(--)	(++)	(--)
	NRP1	(+++)	(--)	(+++)	(--)
	ITGAX	(+++)	(--)	(+++)	(--)
	TBX21	(++)	(--)	(++)	(--)
	STAT4	(++)	(--)	(++)	(--)
Th1	STAT1	(++)	(-)	(++)	(-)
	IFNG	(+)	(-)	(+)	(-)
	TNF	(++)	(-)	(++)	(-)
	GATA3	(+++)	(--)	(++)	(--)
Th2	STAT6				
	STAT5A	(++)	(-)	(++)	(-)
	IL13	(++)	(-)	(+)	(-)
Tfh	BCL6	(+++)	(--)	(+++)	(--)
	STAT3	(++)	(-)	(++)	(-)
Th17	IL17A				
	FOXP3	(+++)	(--)	(+++)	(--)
	CCR8	(+++)	(--)	(+++)	(--)
T reg	STAT5B	(++)		(+)	
	TGFB1	(+++)	(--)	(+++)	(--)
	PDCD1	(++)	(--)	(++)	(--)
	CTLA4	(+++)	(--)	(++)	(--)
T cell exhaustion	LAG3	(++)	(--)	(++)	(--)
	HAVCR2	(+++)	(--)	(+++)	(--)
	GZMB			(+)	
	CD14	(+++)	(--)	(+++)	(--)
	FERMT3	(+++)	(--)	(+++)	(--)
	GPSM3	(+++)	(--)	(+++)	(--)
Myeloid-derived suppressor cells	IL18BP	(+++)	(--)	(+++)	(--)
	PSAP	(+++)	(--)	(+++)	(--)
	PTGES2	(-)		(-)	
	CFD	(+)	(-)	(+)	(-)
	MBL2	(+)	(-)	(+)	(-)
	C2	(+)	(-)	(+)	(-)
	C5	(+)			
Complement	C8G	(-)			
	MASP2				
	C3	(+++)	(--)	(+++)	(--)
	C1S	(+++)	(--)	(+++)	(--)

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
