# Peer review of "Integrated Multi-Omics Investigations of Metalloproteinases in Colon Cancer: Focus on MMP2 and MMP9"

_ijms, 2021, doi:10.3390/ijms222212389_

Round 1
Reviewer 1 Report
Buttacavoli et al., performed analysis from different database platform to investigate the expression of different Metalloproteinases (MMPs) in colon cancer (CRC) patients. The authors also identify the relationship between the MMPs expression and prognosis and correlation between MMPs expression and Immune cells infiltration in CRC patients.
It is a very well written manuscript with clear aims. The methods used in the study are well described. Although the role for individual MMPs in colon cancer have been well described, this manuscript compared the expression of different members of MMPs in the same tumour and normal tissues from different CRC patients. Therefore, the results provide a robust conclusion to strength the relationship between MMPs and CRC.
Major experiments: The expression of different MMPs should also be correlated with different stage of CRC patients.
Author Response
Reviewer 1:
Buttacavoli et al., performed analysis from different database platform to investigate the expression of different Metalloproteinases (MMPs) in colon cancer (CRC) patients. The authors also identify the relationship between the MMPs expression and prognosis and correlation between MMPs expression and Immune cells infiltration in CRC patients.
It is a very well written manuscript with clear aims. The methods used in the study are well described. Although the role for individual MMPs in colon cancer have been well described, this manuscript compared the expression of different members of MMPs in the same tumour and normal tissues from different CRC patients. Therefore, the results provide a robust conclusion to strength the relationship between MMPs and CRC.
Reply: We wish to thanks the reviewer for his pleasures.
Major experiments: The expression of different MMPs should also be correlated with different stage of CRC patients.
Reply: According to the reviewer suggestion we performed the correlation analysis between MMPs expression and cancer stages using UALCAN database and the results were added as supplementary Table 1. The results were explained in the result section, accordingly.
Reviewer 2 Report
I highly appreciate the work. The search for new biomarkers should allow individualized therapeutic management.The manuscript applies to multi-omics investigation of MMPs in colon cancer. The Authors analyzed by data mining 24 MMP genes grouped into 6 subgroups and they used several databases to deeply mine different expression, genetic and epigenetic alteration. Especially, I would like to underline the disscusion - the relationship MMPs with the tumor environment and that the MMPs expression is regulated by inflammatory cells, tumor and stromal cells. Cautious discussion indicates that further studies are needed and the ambiguity of the results does not yet indicate potential relevance in clinical practice. The literature has been carefully selected. The paper is eligible for publication.
Author Response
I highly appreciate the work. The search for new biomarkers should allow individualized therapeutic management.The manuscript applies to multi-omics investigation of MMPs in colon cancer. The Authors analyzed by data mining 24 MMP genes grouped into 6 subgroups and they used several databases to deeply mine different expression, genetic and epigenetic alteration. Especially, I would like to underline the disscusion - the relationship MMPs with the tumor environment and that the MMPs expression is regulated by inflammatory cells, tumor and stromal cells. Cautious discussion indicates that further studies are needed and the ambiguity of the results does not yet indicate potential relevance in clinical practice. The literature has been carefully selected. The paper is eligible for publication.
Reply: We wish to thanks the reviewer for his pleasures.
Round 2
Reviewer 1 Report
Thank you for responding to my question.